# Training and Inference on Any-Order Autoregressive Models the Right Way

**Andy Shih**
Stanford University
andyshih@cs.stanford.edu

**Dorsa Sadigh**
Stanford University
dorsa@cs.stanford.edu

**Stefano Ermon**
Stanford University
ermon@cs.stanford.edu

## Abstract

Conditional inference on arbitrary subsets of variables is a core problem in probabilistic inference with important applications such as masked language modeling and image inpainting. In recent years, the family of Any-Order Autoregressive Models (AO-ARMs) – closely related to popular models such as BERT and XLNet – has shown breakthrough performance in arbitrary conditional tasks across a sweeping range of domains. But, in spite of their success, in this paper we identify significant improvements to be made to previous formulations of AO-ARMs. First, we show that AO-ARMs suffer from redundancy in their probabilistic model, i.e., they define the same distribution in multiple different ways. We alleviate this redundancy by training on a smaller set of univariate conditionals that still maintains support for efficient arbitrary conditional inference. Second, we upweight the training loss for univariate conditionals that are evaluated more frequently during inference. Our method leads to improved performance with no compromises on tractability, giving state-of-the-art likelihoods in arbitrary conditional modeling on text (Text8), image (CIFAR10, ImageNet32), and continuous tabular data domains.

## 1 Introduction

Generative modeling has seen tremendous progress in building highly expressive models [2, 31], but relatively little effort has been put into supporting efficient probabilistic inference. Most existing models with deep neural architectures do not admit efficient inference on conditional queries of the form $p(\boldsymbol{x}_u|\boldsymbol{x}_v)$, where $u$ and $v$ are disjoint subsets of the variables of the joint distribution. Such queries, however, have many important applications such as masked language modeling [42], image inpainting [43], and more. For example, multi-modal models learn a joint distribution over all data modalities, and at test time may only be presented with some subset (unknown in advance) of modalities [41, 20]. In robot shared-autonomy, an operator at test time may choose to provide a subset of the action inputs, leaving the model to fill in the remaining action dimensions [8, 23].

Evidently, expressive generative models that can support efficient conditional inference can bring progress to many applications. Towards this end, the family of Any-Order Autoregressive Models (AO-ARMs) [37, 42, 35, 14] has shown surprising success. AO-ARMs are built on the following insights. An autoregressive model defines an ensemble of univariate conditionals $p(x_t|\boldsymbol{x}_{<t})$ that leads to efficient inference on certain conditional queries: ones whose variables form a prefix of the ordering (Fig. 1a). However, in standard autoregressive models, all other conditionals are defined only implicitly through Bayes' rule, and hence do not admit efficient inference routines. To fix this, consider training another autoregressive model on the reverse ordering, which now defines a larger set of univariate conditionals that can answer queries of either the prefix or the suffix of the ordering (Fig. 1b). AO-ARMs take this insight to the extreme by training on all possible orderings, enabling efficient inference on all possible conditionals. Remarkably, recent efforts in scaling up AO-ARMs

36th Conference on Neural Information Processing Systems (NeurIPS 2022).

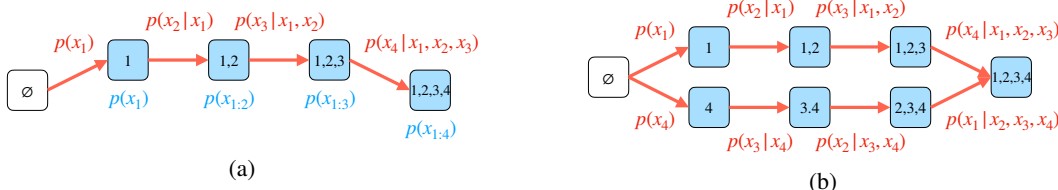

Figure 1: (a) An autoregressive model defines $N$ univariate conditionals $p(x_t|\boldsymbol{x}_{<t})$, allowing for efficient conditional inference when variable subsets form a prefix of the ordering, e.g., $p(\boldsymbol{x}_{1:3})$. (b) Learning an additional autoregressive model on the reverse ordering $x_4, x_3, x_2, x_1$ enables efficient inference on more queries: when variables form suffixes, e.g., $p(\boldsymbol{x}_{2:4})$. However, this leads to redundancy in the probabilistic model, as the joint distribution $p(\boldsymbol{x}_{1:4})$ is now defined in two different ways. This redundancy is exacerbated in AO-ARMs as a result of learning on all possible orders.

have worked extremely well, with breakthrough performance in masked language modeling [42], image inpainting [14], and more [35].

However, in spite of their already strong empirical results, we identify significant improvements that can be made to AO-ARMs. We begin by showing that previous formulations of AO-ARMs suffer from redundancy in their probabilistic model. Consider again the example of training two autoregressive models on the lexicographical and reverse ordering (Fig. 1b). This already unnecessarily defines the same joint distribution in two different ways, as there are two paths to the node $p(\boldsymbol{x}_{1:4})$ corresponding to two distinct factorizations. By scaling up to all possible orderings, AO-ARMs inadvertently model the same distribution with a huge amount of redundancy. Such redundancy makes training inefficient and, much more importantly, leads to worse asymptotic performance. Due to finite capacity of the model, attempting to learn too many univariate conditionals can lead to a poor fit on individual ones.

In this paper, we present **MAC** – Mask-tuned Arbitrary Conditional Models – which proposes two key insights for reducing redundancy and improving model performance. The first insight is that arbitrary conditional inference can still be computed efficiently *without* relying on all possible univariate conditionals. For example in Fig. 1b, omitting the edge $p(x_1|x_2, x_3, x_4)$ still maintains efficient inference for all the nodes in the graph, since there is another path for computing $p(\boldsymbol{x}_{1:4})$. Based on this observation, we reduce redundancy of AO-ARMs by training on a smaller set of univariate conditionals that still supports tractable arbitrary conditional inference. The second insight is that some univariate conditionals are evaluated much more often than others during inference. Therefore, we upweight the training loss for the more frequently occurring univariate conditionals, thereby aligning the training and inference objectives more closely.

Combining these insights, MAC leads to a strictly-improved formulation of AO-ARMs that, amazingly, gives better arbitrary conditional and joint likelihoods with *no compromises* on tractability. We achieve state-of-the-art likelihoods for arbitrary conditional modeling across multiple domains such as text (Text8), image (CIFAR10, ImageNet32), and continuous tabular data. Finally, we conclude by demonstrating a novel application of AO-ARMs to robot shared-autonomy.

## 2   Background

**Problem definition**   Marginal and conditional inference are important queries for handling partial evidence on arbitrary subsets of variables, which is relevant for tasks such as masked language modeling or image inpainting. We let a **mask** $e$ be a subset of variables, and its **cardinality** $|e|$ be the number of elements in the subset. A masked input $\boldsymbol{x}_e$ is a partial instantiation on the variable subset $e$ using the values taken on by input $\boldsymbol{x}$. Marginal inference refers to queries of the form $p(\boldsymbol{x}_e) = \int p(\boldsymbol{x}_e \boldsymbol{x}_q) d\boldsymbol{x}_q$, where $q = X \setminus e$ and $\boldsymbol{x}_e \boldsymbol{x}_q$ denotes the union of the partial instantiations. Conditional inference queries $p(\boldsymbol{x}_u|\boldsymbol{x}_e) = p(\boldsymbol{x}_u \boldsymbol{x}_e)/p(\boldsymbol{x}_e)$ are similar and can be computed as a ratio of two marginals, so we will refer to marginals and conditionals interchangeably. To evaluate model performance on marginal inference queries over a test dataset $X_{\text{test}}$, we assume a test mask distribution $M$ over the $2^N$ possible masks. We sample masks $e$ from $M$ independently from data,

and use the negative marginal log-likelihood of the masked input $\boldsymbol{x}_e$ as the loss.

$$\mathcal{L}_M = - \sum_{\boldsymbol{x} \in X_{\text{test}}} \mathbb{E}_{e \sim M} \log p(\boldsymbol{x}_e) \tag{1}$$

Computing arbitrary marginals and conditionals has been a long-standing challenge in probabilistic inference, with a rich history of techniques [28, 26, 15, 5, 40] ranging from belief propagation, variational inference, to MCMC. Such queries are inherently difficult in high-dimensions: given access to joint likelihoods $p(\boldsymbol{x})$, naïve computation of marginal likelihoods $p(\boldsymbol{x}_e)$ requires integration over the missing $N - |e|$ dimensions. As a result, most traditional approaches model the joint distribution using non-neural architectures that admit efficient inference but are less expressive [18].

**Autoregressive Models**   Autoregressive models are an influential family of generative models that represents complex high-dimensional distributions by modeling a single dimension at a time. They parameterize a joint distribution $p(\boldsymbol{x})$ over $N$ dimensions by factorizing it into univariate conditionals $\prod_{t=1}^{N} p(x_{\sigma(t)} | \boldsymbol{x}_{\sigma(<t)}; \sigma)$ via chain rule (Fig. 1a), using an ordering $\sigma$ of the $N$ variables. We write $\sigma(t)$ and $\sigma(< t)$ to denote the masks corresponding to the $t$-th element and the first $t - 1$ elements of the ordering, respectively.

The univariate conditionals can be learned via a weight-tied neural network, and composing them together leads to highly expressive architectures in practice, such as PixelCNN [38, 32] or Transformers [39]. Autoregressive models support efficient inference on the joint distribution, and on a specific type of marginal query: ones where the mask forms a prefix of the variable order, i.e., $\sigma(\leq |e|) = e$ (Fig. 1a). We will call such a mask $e$ and ordering $\sigma$ *compatible*.

However, autoregressive models cannot support efficient arbitrary marginal inference in general. Next, we present AO-ARMs, which extend autoregressive models in a way that does support arbitrary marginal inference.

## 2.1   Any-Order Autoregressive Models (AO-ARMs)

AO-ARMs [37, 35, 14] learn a single model that can generate the joint distribution autoregressively using any ordering of the $N$ variables, by modeling $\prod_{t=1}^{N} p(x_{\sigma(t)} | \boldsymbol{x}_{\sigma(<t)}; \sigma)$ for all orderings $\sigma$. Even though there are $N!$ different orderings, their chain-rule factorization is built from univariate conditionals, of which there are "only" $N2^{N-1}$. Therefore, an AO-ARM is a model that defines the $N2^{N-1}$ distinct univariate conditionals $p(x_j | \boldsymbol{x}_{e \setminus j})$, where $j \in e$.

To answer a marginal inference query $p(\boldsymbol{x}_e)$ with an AO-ARM, we choose an order $\sigma$ that is compatible with $e$, so that $\sigma(\leq |e|) = e$. Then, we simply evaluate each of the univariate conditionals $p(\boldsymbol{x}_e) = \prod_{t=1}^{|e|} p(x_{\sigma(t)} | \boldsymbol{x}_{\sigma(<t)}; \sigma)$ in the autoregressive factorization of $\boldsymbol{x}_e$.

**Training AO-ARMs**   Architecturally, an AO-ARM models all univariate conditionals via a weight-tied neural network, by using a special (so-called "absorbing-state" [1, 14]) token for variables that are not present in the evidence set. To enable parallel optimization, the AO-ARM architecture is designed so that given evidence $\boldsymbol{x}_e$, the model can predict the univariate conditionals $p(x_i | \boldsymbol{x}_e)$ for all $i \in X \setminus e$ at once. Importantly, this parallelization works on non-causal architectures, opening the doors to architectures similar in flexibility to diffusion models [33, 34, 13].

In previous works [37, 35, 14], AO-ARMs are trained to maximize the joint likelihood of a datapoint $\boldsymbol{x}$ under the expectation over the uniform distribution $\mathcal{U}_\sigma$ of orders.

$$\log p(\boldsymbol{x}) = \log \mathbb{E}_{\sigma \sim \mathcal{U}_\sigma} \sum_{t=1}^{N} p(x_{\sigma(t)} | \boldsymbol{x}_{\sigma(<t)}; \sigma) \geq \mathbb{E}_{\sigma \sim \mathcal{U}_\sigma} \sum_{t=1}^{N} \log p(x_{\sigma(t)} | \boldsymbol{x}_{\sigma(<t)}; \sigma) \tag{2}$$

This objective can be simplified by treating $t$ as a random variable with a uniform distribution $\mathcal{U}_t$ over cardinalities 1 to $N$. We let $M_{\text{card-edge}}$ be the distribution over the tuple $(\sigma(t), \sigma(< t))$ where $\sigma \sim \mathcal{U}_\sigma$ and $t \sim \mathcal{U}_t$, giving the following loss function for an AO-ARM parameterized by $\theta$.

$$\mathcal{L}(\theta) = -\mathbb{E}_{\sigma \sim \mathcal{U}_\sigma} \sum_{t=1}^{N} \log p_\theta(x_{\sigma(t)} | \boldsymbol{x}_{\sigma(<t)}; \sigma) = -N \cdot \mathbb{E}_{i, e \sim M_{\text{card-edge}}} \log p_\theta(x_i | \boldsymbol{x}_e) \tag{3}$$

In practice, due to parallel optimization, we write the objective in the following equivalent form.

$$\mathcal{L}(\theta) = -N \cdot \mathbb{E}_{-,e \sim M_{\text{card-edge}}} \frac{1}{N - |e|} \sum_{i \in X \setminus e} \log p_\theta(x_i | \boldsymbol{x}_e) \tag{4}$$

Compared to other arbitrary conditional models, AO-ARMs consistently give state-of-the-art performance on benchmarks across a range of domains [37, 35, 14]. Besides good empirical performance, they also serve as a conceptual unification between autoregressive models and diffusion models [14, 1], showing promise for both practical and theoretical advancements.

## 3 Improving Any-Order Autoregressive Models with MAC

Despite the success of AO-ARMs, in this section we identify two key deficiencies in previous formulations of AO-ARMs. We propose solutions to them by re-interpreting AO-ARMs from the perspective of recursive decomposition over marginals. Our resulting method MAC leads to consistent improvements and state-of-the-art likelihoods across a range of domains in Sec. 4.

**(A)** The first deficiency is that of redundancy. While being order-agnostic is exactly what enables arbitrary conditionals, learning the same distribution with multiple orderings makes training inefficient. More importantly, since the model has limited capacity (especially with the weight-tied architecture), learning orderings redundantly hurts asymptotic performance.

**(B)** The second deficiency stems from the AO-ARM training objective in Eq. 2. This objective focuses only on the joint likelihood, leading to a mismatch with the marginal likelihood evaluation objective in Eq. 1, even though computing marginals is the key feature of AO-ARMs.

When viewing AO-ARMs as autoregressive models that generate using all orders, it is not obvious how to fix the above two deficiencies. For example, if we try to omit certain orders to address **(A)**, we may lose the ability to efficiently answer marginals, e.g., queries that are prefixes of the orders we removed. Similarly for **(B)**, it is unclear how to choose a distribution of orderings to best account for arbitrary conditional inference. To solve these issues, we re-interpret AO-ARMs from the perspective of computing arbitrary conditional probabilities via recursive decomposition over masks.

### 3.1 Re-interpreting AO-ARMs as recursive decomposition on a binary lattice

A mask $e$ is associated with the set of marginal inference queries of the form $p(\boldsymbol{x}_e)$. We will refer to this set of queries as the corresponding **task** for mask $e$. We represent these masks/tasks as nodes in Fig. 1. Supporting efficient computation of each task, therefore, implies efficient arbitrary marginal inference. However, for masks with high cardinality, the corresponding task involves learning a high-dimensional distribution.

Fortunately, these masks do not actually form standalone tasks, but are subproblems of one another. In particular, given non-empty mask $e$ and a single variable $j \in e$, we can write $p(\boldsymbol{x}_e)$ as $p(\boldsymbol{x}_{e \setminus j})p(x_j | \boldsymbol{x}_{e \setminus j})$. Then, we delegate the computation of $p(\boldsymbol{x}_{e \setminus j})$ to the task associated with mask $e \setminus j$, and are only left with estimating the univariate conditional $p(x_j | \boldsymbol{x}_{e \setminus j})$.

From this perspective, we can view AO-ARMs as solving an ensemble of intertwined tasks, one for each of the $2^N$ possible masks. We visualize this as a binary lattice in Fig. 2a over $N = 4$ variables, where we have $2^4 = 16$ nodes representing the possible masks/tasks, with an edge connecting nodes $e'$ and $e$ if $e' = e \setminus j$ for some singleton variable $j \in e$. To answer a marginal query $p(\boldsymbol{x}_e)$, we choose an element $j \in e$ and compute $p(\boldsymbol{x}_e) = p(\boldsymbol{x}_{e \setminus j})p(x_j | \boldsymbol{x}_{e \setminus j})$, which corresponds to moving along the edge from node $e$ to node $e \setminus j$ in the lattice. We proceed recursively on $p(\boldsymbol{x}_{e \setminus j})$ until we reach $p(\boldsymbol{x}_\emptyset) = 1$.

Under this recursive decomposition framework, two key concepts are the decomposition protocol, which dictate how edges are chosen, and the mask/edge distributions, which specify the probability that different marginals and univariate conditionals are evaluated.

**Definition 1** *A **decomposition protocol** $w$ defines, for each non-empty mask $e$, a probability distribution $w_e$ over the elements in $e$. In other words, $\sum_{j \in e} w_e(j) = 1$ and $w_e(j) \geq 0$ for $e \neq \emptyset$.*

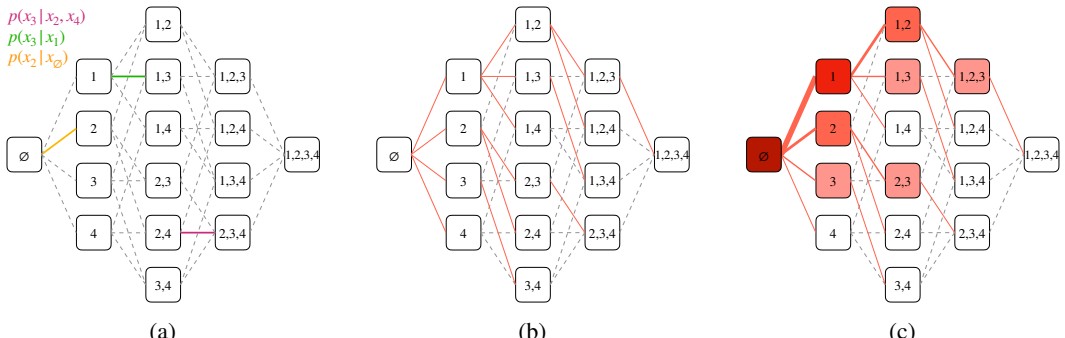

Figure 2: (a) For $N = 4$, marginal queries can take on one of $2^4 = 16$ possible masks, represented as square nodes. These masks are related to one another through univariate conditionals (edges), since $p(x_e) = p(\boldsymbol{x}_{e\backslash j})p(x_j|\boldsymbol{x}_{e\backslash j})$. For example, given $p(x_1)$, to compute $p(\boldsymbol{x}_{1,3})$ we only need to learn the univariate conditional $p(x_3|x_1)$, shown as the green edge. (b) We don't need to learn all univariate conditionals (edges) to compute arbitrary marginals (nodes). It suffices to learn one left-going edge for every node besides the $\emptyset$ node. (c) Given a marginal query at a node, we recursively decompose it by following red edges until we reach the $\emptyset$ node. Hence, some edges / nodes will be traversed more often, so we should train on them more often.

A decomposition protocol $w$ specifies a process for answering marginal inference queries. Given a marginal query at node $e$, we sample an element $j \sim w_e$ and compute $p(\boldsymbol{x}_e) = p(\boldsymbol{x}_{e\backslash j})p(x_j|\boldsymbol{x}_{e\backslash j})$. We proceed recursively on $p(\boldsymbol{x}_{e\backslash j})$ until we reach $p(\boldsymbol{x}_\emptyset) = 1$.

**Definition 2** *A **mask distribution** is a probability distribution over the $2^N$ possible variable subsets for marginals. An **edge distribution** is a probability distribution over the $N2^{N-1}$ possible variable subsets for univariate conditionals.*

Mask and edge distributions are relevant for specifying the *test mask distribution* (Eq. 1) of the problem domain, and the *training edge distribution* for the AO-ARM. When viewed under this recursive decomposition interpretation, previous formulations of AO-ARMs use a random decomposition protocol $w$-RND where $w\text{-RND}_e(j) = 1/|e|$. Moreover, they use a training edge distribution $M_{\text{card-edge}}$ (Eq. 3), which unfortunately neglects to take into account the test mask distribution $M$.

### 3.2 MAC: Mask-tuned Arbitrary Conditional Model

We can now describe our method MAC. MAC improves upon previous AO-ARMs by choosing a custom decomposition protocol and deriving a training edge distribution that aligns with the test mask distribution $M$.

**(A) Reducing edge redundancy**  To answer arbitrary marginals, we don't need to learn to generate using all possible orderings. Rather, it suffices to decompose each task into any one of its children tasks. This constraint corresponds to learning a single (red) incoming edge for each node besides the $\emptyset$ node. We show an example in Fig. 2b, where we connect each node to its leftward parent obtained by removing the greatest element of the node's mask. This is precisely the decomposition protocol that we use for MAC: $w\text{-MAC}_e(j) = 1$ if $j$ is the greatest element in $e$.

**(B) Matching edge distribution**  When presented with a marginal query $e$, the decomposition protocol $w$ traverses a path from node $e$ to node $\emptyset$, evaluating the univariate conditionals along the way. Simulating $w$ on masks $e \sim M$ leads to an *induced edge distribution* $D_{M,w}$ that represents how frequently different univariate conditionals are evaluated during the inference process, which we depict in Fig. 2c using red edges with different thicknesses. We can see, for example, that under the protocol $w$-MAC, univariate conditionals such as $p(x_1|\boldsymbol{x}_\emptyset)$ are evaluated highly frequently because it appears as a subproblem to many marginal queries. Therefore, we should tune our training distribution on univariate conditionals to match this induced edge distribution $D_{M,w}$ given by the downstream test mask distribution $M$ and the decomposition protocol $w$-MAC.

Abstractly, let $\pi \sim w(e)$ denote the probability that $w$ chooses a path of edges $\pi = \{(i_t, e_t)\}_{t=1}^{|e|}$ when decomposing $e$. In particular, $e_1 = \emptyset$, and $i_t$ is a variable in $e \setminus e_t$, and $e_{t+1} = i_t \cup e_t$. Then we can write the marginal inference objective as follows, where $C$ is the constant $\mathbb{E}_{i,e \sim D_{M,w}}|e|$.

$$\mathbb{E}_{e \sim M}[\log p(\boldsymbol{x}_e)] = \mathbb{E}_{e \sim M} \mathbb{E}_{\pi \sim w(e)} \sum_{t=1}^{|e|} \log p(x_{i_t}|\boldsymbol{x}_{e_t}) \tag{5}$$

$$= C \cdot \mathbb{E}_{i,e \sim D_{M,w}} \log p(x_i|\boldsymbol{x}_e) \tag{6}$$

Eq. 5 leads to the following algorithms for training MAC (Alg. 1). We sample a marginal query $p(\boldsymbol{x}_e)$ by independently sampling a mask $e \sim M$ and an input $\boldsymbol{x} \sim X$. Then, we simulate the path taken on the lattice by the decomposition protocol $w$ by iteratively sampling an element $j \sim w_e$ and updating $e$ to be $e \setminus j$. Throughout this process, we optimize our model on the edges (univariate conditionals) traversed on this path. During testing (Alg. 2), we similarly step through the decomposition protocol and add together the univariate conditional likelihoods along the path.

| **Algorithm 1:** Training MAC |
|---|
| **Input:** Test mask distribution $M$, decomposition protocol $w$, training data distribution $X$, model $\theta$ |
| 1 **while** *training* **do** |
| 2    $e \sim M, \boldsymbol{x} \sim X$ |
| 3    **while** $e \neq \emptyset$ **do** |
| 4      $j \sim w_e$ |
| 5      $\theta \leftarrow \theta + \nabla_\theta \log p_\theta(x_j|\boldsymbol{x}_{e \setminus j})$ |
| 6      $e \leftarrow e \setminus j$ |

| **Algorithm 2:** Testing MAC |
|---|
| **Input:** Mask $e$, decomposition protocol $w$, test data $\boldsymbol{x}$, model $\theta$ |
| **Output:** Marginal log-likelihood $\log p_\theta(\boldsymbol{x}_e)$ |
| 1   $r \leftarrow 0$ |
| 2 **while** $e \neq \emptyset$ **do** |
| 3    $j \sim w_e$ |
| 4    $r \leftarrow r + \log p_\theta(x_j|\boldsymbol{x}_{e \setminus j})$ |
| 5    $e \leftarrow e \setminus j$ |
| **Return:** $r$ |

In practice, since the decomposition protocol $w$-MAC deterministically removes the greatest element, there is an efficient way to directly sample from $D_{M,w\text{-MAC}}$ in batch (Eq. 6), without having to simulate $w$-MAC as in Alg. 1. We describe these details in the Appendix. Finally, to conclude this section, we discuss two important design choices and practical considerations for $w$ and $D_{M,w}$.

**Parallel training** As noted in Section 2, the architecture of AO-ARMs enables training on univariate conditionals of the form $p(x_i|\boldsymbol{x}_e)$ for $i \in X \setminus e$ in parallel. Therefore, rather than dealing with a training distribution over univariate conditionals $D_{M,w}(i, e)$, we instead work with one over masks $D_{M,w}(e)$. To a close approximation, we will simply let $D_{M,w}(e) \propto \sum_i D_{M,w}(i, e)$, visualized as shaded nodes in Fig. 2c.

**Cardinality Reweighting (heuristic)** We present one additional technique for tuning the training mask distribution, with ablations and further discussion in later sections. To foster generalization of our weight-tied neural network, we reweigh the probability of a mask based on its cardinality $e$, by using a final training distribution of $D_{M,w}^{\text{CR}}(e) \propto (1 + |e|)D_{M,w}(e)$, which we sample using SIR (Sampling-Importance-Resampling).

With parallelism and cardinality reweighting, the final objective for MAC is the following:

$$\mathcal{L}_M(\theta) := -\mathbb{E}_{e \sim D_{M,w\text{-MAC}}^{\text{CR}}} \frac{1}{N - |e|} \sum_{i \in X \setminus e} \log p_\theta(x_i|\boldsymbol{x}_e) \tag{7}$$

The decomposition protocol $w$-MAC recursively removes the greatest element of a mask, where comparison between two elements $j_1 > j_2$ is determined by a global canonical ordering (e.g. lexicographical ordering).

## 4 Experiments

We evaluate MAC on high-dimensional language and image domains, and on a set of continuous tabular benchmarks. We focus on two metrics: joint likelihood and marginal likelihood of the test set. On both metrics, MAC shows state-of-the-art performance among arbitrary conditional models on the

majority of benchmarks. We conclude by demonstrating a novel application of arbitrary conditional models on a simulated robot shared-autonomy task, where we learn a conditional policy in a 9-DoF action space through implicit behavioral cloning with MAC.

**Note on computation**   Each run was done on a single NVIDIA A40. For the language and image experiments, we trained each model for approximately two weeks. Although this was not enough to match the total number of epochs trained by the baseline ARDM[1], we were still able to show state-of-the-art performance for arbitrary conditional models on 2 out of the 3 language/image benchmarks, and beat the baselines on all 3 benchmarks when compared under the same number of training epochs.

Table 1: Character-level modeling of the Text8 dataset (without additional context), in bpd.

|  | joint | marginal |
|---|---|---|
| **Joint Models** | | |
| *from literature* | | |
| Transformer [39] | 1.35 | − |
| **Arb. Cond. Models** | | |
| *from literature* | | |
| OA-Transformer [14, 42] | 1.64 | − |
| D3PM [1] | 1.47 | − |
| ARDM [14] (14000 epochs) | 1.43 | − |
| *our experiments* | | |
| ARDM (3000 epochs) | 1.48 | 1.12 |
| MAC (3000 epochs) | 1.40 | 1.09 |

Figure 3: Ablation on choices of training mask distribution. The best performing mask distribution $D_{M,w\text{-MAC}}^{\text{CR}}$ uses the $w$-MAC decomposition protocol with cardinality reweighing.

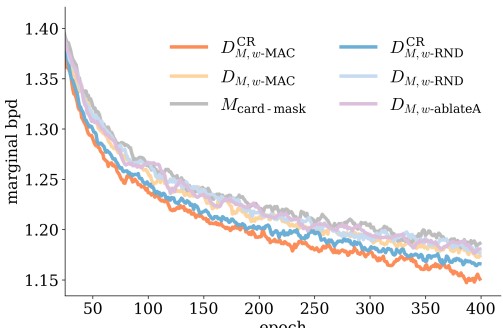

## 4.1   Language

We learn a character-level model on chunks of 250 characters using the Text8 dataset [24], which consists of 100M characters. We follow the setup from previous work in modeling the chunked text segments *without* any additional context, and use the same 12-layer Transformer architecture as ARDM and D3PM [14, 1]. We report the joint and marginal likelihoods in terms of bpd (lower is better), where the marginal bpd is computed w.r.t. to the test mask distribution $M_{\text{card-mask}}(e) \propto 1/\binom{N}{|e|}$ as used in [35][2]. In Tab. 1 we see that MAC outperforms all existing arbitrary conditional models in joint likelihood and comes close to a standard Transformer trained only to model in one variable order, even when trained only on a fraction of the total number of epochs by baselines in literature. Similarly, MAC also gets better marginal bpd compared to ARDM.

**Ablations**   In Fig. 3 we present ablation studies on different choices of the training mask distribution. The baseline method (ARDM) trains on the $M_{\text{card-mask}}$ mask distribution (grey). Using only insight **(A)** from Sec. 3 with a deterministic decomposition protocol but no edge-weighting gives slight improvements $D_{M,w\text{-ablateA}}$ (light purple). Using only insight **(B)** from Sec. 3 with edge-weighting but random decomposition protocol also leads to slight improvements $D_{M,w\text{-RND}}$ (light blue). Using both insights **(A)** and **(B)** gives the $w$-MAC protocol which leads to more improvements $D_{M,w\text{-MAC}}$ (light orange). Lastly, the cardinality reweighing heuristic improves both methods even more (dark blue/orange). The best performance is given by $D_{M,w\text{-MAC}}^{\text{CR}}$ in dark orange, which is the setting we use for MAC in all our other experiments.

## 4.2   Images

We evaluate MAC on CIFAR10 [19] and ImageNet32 [4, 6], both of which consist of images of dimension 3072. Again, we follow all experimental settings from previous work [1, 14, 17], using a U-Net with 32 ResNet Blocks interleaved with attention layers, and use rotation/flip data augmentation. In Tab. 2&3 we see that MAC outperforms ARDM on joint bpd and marginal bpd, evaluated on the

---

[1]Previous works were unable to share checkpoints of baseline methods.

[2]$M_{\text{card-mask}}$ effectively samples a cardinality $c$, and then samples uniformly among masks with cardinality $c$.

$M_\text{card-mask}$ mask distribution. Surprisingly, MAC even gives the best joint bpd on ImageNet32 when compared to joint models that do not support arbitrary conditionals.

Table 2: Pixel modeling of the ImageNet32 dataset (with no data augmentation), in bpd.

|  | joint | marginal |
|---|---|---|
| **Joint Models** | | |
| *from literature* | | |
| Image Transformer [27] | 3.77 | – |
| VDM [17] | 3.72 | – |
| **Arb. Cond. Models** | | |
| *our experiments* | | |
| ARDM (16 epochs) | 3.60 | 2.10 |
| MAC (16 epochs) | 3.58 | 2.08 |

Table 3: Pixel modeling of the CIFAR10 dataset (with rotation/flip data augmentation), in bpd.

|  | joint | marginal |
|---|---|---|
| **Joint Models** | | |
| *from literature* | | |
| PixelCNN++ [32] | 2.88 | |
| Sparse Transformer [3, 16] | 2.56 | – |
| VDM [17] | 2.49 | – |
| **Arb. Cond. Models** | | |
| *from literature* | | |
| D3PM [1] | 3.44 | – |
| ARDM [14] (3000 epochs) | 2.69 | – |
| *our experiments* | | |
| ARDM (1200 epochs) | 2.86 | 1.84 |
| MAC (1200 epochs) | 2.81 | 1.81 |

## 4.3 Continuous Tabular Data

Next, we consider a tabular domain with variables taking on continuous values [35]. The main challenge with continuous values for AO-ARMs is the parameterization of the univariate conditionals, since 1-D Gaussians or even mixtures of Gaussians have limited expressiveness. To this end, ACE [35] proposes to parameterize the 1-D conditionals as EBMs. We build MAC on top of ACE, keeping all hyperparameters and experimental setup the same, modifying only the training mask distribution. In Tab. 4&5, we see that MAC gives better marginal and conditional likelihoods than ACE on most settings, and outperforms other arbitrary conditional models such as SPNs [30, 25] and ACFlow [22].

Table 4: Marginal log-likelihood on 5 continuous tabular benchmarks (higher is better). The mask cardinality settings kept consistent with the ones reported in [35].

|  | power | | | gas | | | hepmass | | | miniboone | | | bsds | | |
|---|---|---|---|---|---|---|---|---|---|---|---|---|---|---|---|
| Mask cardinality | 3 | 5 | 6 | 3 | 5 | 8 | 3 | 5 | 10 | 3 | 5 | 10 | 3 | 5 | 10 |
| SPFlow [25] | -0.63 | 1.01 | -0.12 | 0.68 | 1.88 | 4.81 | -4.01 | -6.58 | -13.38 | -2.21 | -4.31 | -9.85 | -2.87 | -4.42 | -8.15 |
| ACFlow [22] | -0.57 | 1.34 | 0.42 | 0.78 | 3.01 | 10.13 | -4.03 | -6.19 | -11.58 | -2.76 | -5.31 | -10.36 | 5.06 | 9.26 | 19.60 |
| ACE [35] | -0.56 | 1.42 | 0.58 | 1.31 | 4.31 | 12.20 | **-4.00** | -5.91 | -10.72 | -2.13 | -3.80 | -7.94 | **5.10** | **9.37** | 20.31 |
| MAC | **-0.55** | **1.43** | **0.61** | **1.59** | **4.98** | **13.02** | **-4.00** | **-5.90** | **-10.69** | **-2.12** | **-3.76** | **-7.76** | **5.10** | **9.37** | **20.33** |

Table 5: Conditional log-likelihood on 5 continuous tabular benchmarks.

|  | power | gas | hepmass | miniboone | bsds |
|---|---|---|---|---|---|
| SPFlow [25] | -1.03 | 4.30 | -12.78 | -18.34 | -24.15 |
| ACFlow [22] | 0.56 | 8.09 | -8.20 | -0.97 | 81.83 |
| ACE [35] | 0.63 | 9.64 | -3.86 | **0.31** | **86.70** |
| MAC | **0.65** | **9.77** | **-3.05** | 0.07 | 86.05 |

Table 6: Shared autonomy on FrankaKitchen with BC operator policy. Full autonomy baseline (IBC) [10]: $2.15 \pm 0.06$

| Conditional Policy | Reward |
|---|---|
| BC | $1.81 \pm 0.08$ |
| MAC | $\mathbf{2.00 \pm 0.05}$ |

## 4.4 Shared Autonomy on FrankaKitchen

Finally, we demonstrate an application of arbitrary conditional models on the task of shared autonomy in robotic manipulation. In shared autonomy settings, an operator aims to control a policy in a complicated action space, but is allowed to delegate partial control to an AI model for easier manipulation. We focus on the simulated FrankaKitchen environment [12, 11], where the goal is to control a 9-DoF robotic arm to move objects around a kitchen.

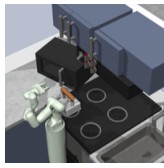

We train a base policy with behavioral cloning (BC) [29] and an arbitrary conditional policy with MAC to solve the `Kitchen-mixed0` task. To simulate shared autonomy, we evaluate

a hybrid policy where at each state, the operator policy (BC) outputs 4 out of the 9 dimensions of the action space, and a conditional policy fills in the remaining 5 action dimensions. In Tab. 6 we see that conditioning using MAC improves shared autonomy as compared to conditioning using an independent BC policy, and comes close to the full autonomy performance [10].

## 5  Discussion

**Sharpen mask distribution**  Intuitively, we should choose a protocol $w$ such that $D_{M,w}$ has low entropy so that our model has a greater chance of revisiting similar masks. To investigate the differences in choices of decomposition protocol, in Tab. 7 we present Monte-Carlo simulations of the induced mask distributions with respect to the test mask distribution $M_{\text{card-mask}}$. As expected, using $w$-MAC leads to lower empirical entropy than using $w$-RND.

Table 7: Monte-Carlo simulation of induced mask distributions for $N = 12$, taking 1e6 samples. For brevity we only display a few selected rows.

| Mask | $M$ | $D_{M,w\text{-RND}}$ | $D_{M,w\text{-MAC}}$ | $D_{M,w\text{-MAC}}^{\text{CR}}$ |
|---|---|---|---|---|
| 011111111110 | 1261 | 222 | 650 | 1827 |
| 100000000000 | 7037 | 14803 | 76381 | 41441 |
| 100000000101 | 407 | 521 | 0 | 0 |
| 100000001000 | 1288 | 1975 | 536 | 430 |
| 111111111110 | 6956 | 433 | 6935 | 21094 |
| Entropy | 7.21 | 5.74 | 4.44 | 5.51 |

**Mask generalization (and why we do Cardinality Reweighting)**  The success of AO-ARMs depends on generalization to unseen masks, since there are exponentially many masks. Training on many masks improves generalization, and can be viewed as a form of data augmentation that allows AO-ARMs to sometimes even outperform standard autoregressive models on joint likelihoods.

In light of this, a decomposition protocol $w$ that skews the mask distribution may have the unfortunate side-effect of hurting generalization. In particular, the mask distribution $D_{M,w\text{-MAC}}$ puts much more weight on low-cardinality masks. This is problematic since training on cardinality-1 masks is not conducive to generalization: there are only $N$ possible cardinality-1 masks, whereas higher cardinalities have exponentially more masks. As such, we proposed $D_{M,w\text{-MAC}}^{\text{CR}}$ (Tab. 7) with the hypothesis that focusing less on low-cardinality masks will lead to better generalization. Our ablation study supports this hypothesis, as cardinality reweighting indeed improves performance (Fig. 3).

**Parallel training**  To account for parallel training, we proposed to train on the mask distribution $D_{M,w}(e) \propto \sum_{i \in X \setminus e} D_{M,w}(i, e)$. However, each pass of $D_{M,w}(e)$ trains on all of $\{p(x_i | \boldsymbol{x}_e) : i \in X \setminus e\}$, which overtrains neighboring edges and leads to a minor distributional mismatch. Aligning this training distribution more closely may lead to improvements, but requires better techniques for parallelization and sampling of masks. We leave this for future investigation.

### 5.1  Related Work

The idea of training AO-ARMs as autoregressive models on all possible orderings was first introduced in NADE [37, 36], and has been seen more recently in models such as ARDM [14] for image / text / audio and ACE [35] for continuous domains. AO-ARMs are also closely related to non-autoregressive language models such as BERT [7] and XLNet [42]. Our method, MAC, proposes improved training and inference techniques for AO-ARMs.

Other techniques have also been proposed for non-autoregressive generation. For example, IN-TRUS [9] models sequences as consecutive insertion operations, as opposed to AO-ARMs which model sequences as consecutive unmasking operations. This also enables arbitrary order of generation, but prevents INTRUS from computing marginal likelihoods on subsets of variables. Another work [44] casts the problem of non-autoregressive generation as a GFlowNet, taking on a reinforcement learning framework with states, actions, and rewards. However, this approach currently does not scale as well as AO-ARMs.

Lastly, some works have studied the problem of learning a single generation order [21]. Although learning orderings is not applicable to vanilla AO-ARMs (since they train on all possible orders uniformly), it is applicable to MAC, which uses decomposition protocols that can be learned. For example, we can learn different canonical orderings of the variable indices in the binary lattice. Though we did not explore this direction thoroughly, we noticed that a strided canonical ordering (0, 32, 64, ..., 1, 33, ...) often did better than the standard lexicographical ordering for images, suggesting promise for learning even better canonical orderings, or learning better protocols in general.

## 5.2 Limitations

In practice, likelihood evaluations of standard AO-ARMs are done by sampling a finite number of orders (typically just a single order), leading to unbiased but approximate estimates of joint likelihoods. Moreover, evaluating marginal $p(\boldsymbol{x}_e)$ and conditional likelihoods $p(\boldsymbol{x}_u|\boldsymbol{x}_e)$ is trickier and has additional tradeoffs. Since AO-ARMs evaluate marginal and conditional likelihoods using only compatible orderings, they are implicitly discarding away all the models in the $N!$ ensemble with incompatible orderings. This can give biased estimates for marginal and conditional likelihoods.

For conditional likelihoods specifically, there are two potential methods for evaluation. First, we can evaluate $\log p(\boldsymbol{x}_u|\boldsymbol{x}_e)$ as $\log p(\boldsymbol{x}_u\boldsymbol{x}_e) - \log p(\boldsymbol{x}_e)$ and estimate two marginal likelihoods. However, since the marginal likelihoods are approximate, this can lead to invalid conditional probability estimates where $p(\boldsymbol{x}_u|\boldsymbol{x}_e) > 1$ (for discrete domains). Second, we can decompose $\boldsymbol{x}_u$ directly and evaluate $\log p(\boldsymbol{x}_u|\boldsymbol{x}_e)$ as $\sum_{i=1}^{|u|} \log p(\boldsymbol{x}_{u_{:i}}|\boldsymbol{x}_{u_{i:}}\boldsymbol{x}_e)$, where the ordering of variables within $\boldsymbol{x}_u$ is sampled randomly. This corresponds to evaluating random path that goes from node $e$ to node $u \cup e$ in the binary lattice. Though this leads to valid probability estimates, the estimates may be more biased because paths that go to node $u \cup e$ without passing through node $e$ are not considered.

MAC shares similar limitations as standard AO-ARMs, where marginal likelihoods estimates may be biased. Evaluating conditional likelihoods as two marginal likelihoods can lead to invalid (greater than 1) estimates, and evaluating them by tracing paths through the lattice from node $e$ to node $u \cup e$ may be more biased.

Nonetheless, both marginal and conditional likelihood estimates (using Method 2) for MAC and standard AO-ARMs *are valid* in the sense that $\sum_{\boldsymbol{x}_u} p(\boldsymbol{x}_u) = 1$ and $\sum_{\boldsymbol{x}_u} p(\boldsymbol{x}_u|\boldsymbol{x}_e) = 1$ for any $u$ and $e$. Training MAC to optimize for marginal likelihoods (Tab. 1,2,3) or for both marginal/conditional likelihoods (Tab. 4,5) still gives good empirical performance despite these limitations.

## 6 Conclusion

We present MAC, an improved method of training and inference on AO-ARMs. MAC trains on a carefully designed distribution of univariate conditionals that **(A)** reduces modeling redundancy of AO-ARMs and **(B)** aligns the training objective of AO-ARMs with arbitrary conditional queries. Our method leads to better joint and arbitrary conditional likelihoods with no sacrifices on tractability. We show state-of-the-art results in arbitrary conditional modeling on text, image, continuous data domains, and present a novel application of arbitrary conditional models to robot shared-autonomy.

## 7 Acknowledgments

We thank Rui Shu, Jiaming Song, Jesse Mu, and anonymous reviewers for their constructive feedback. This research was supported by NSF(#1651565), AFOSR (FA9550-19-1-0024), ARO (W911NF-21-1-0125), ONR, DOE, CZ Biohub, and Sloan Fellowship.

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
