# A Connections between Mask Distributions and Variable Ordering

We draw connections between our proposed framework over mask distributions and prior works' formulation over variable order distributions. In our framework, the decomposition protocol $w$ specifies the probability of reducing a mask into one of its parents. By recursively applying $w$, we can trace a path on the binary lattice from a mask $e$ to the root node $\emptyset$. This path corresponds to a (partial) variable order compatible with $e$. Under this interpretation, since our proposed protocol $w$-MAC is deterministic, it use a deterministic variable ordering (that is different) for each mask.

We can treat the variable order $\sigma$ as a random variable and view the likelihood as a variational objective, as is done in prior work [37, 14]. Previous AO-ARMs, which sample ordering at random, use a uniform prior $\mathcal{U}(\sigma)$ over orderings, leading to the following ELBO.

$$\log p(\boldsymbol{x}) = \log \sum_\sigma p(\sigma)p(\boldsymbol{x}|\sigma) = \log \sum_\sigma \mathcal{U}(\sigma) \prod_{t=1}^N p(x_{\sigma(t)}|\boldsymbol{x}_{\sigma(<t)};\sigma)$$

$$\geq \sum_\sigma \mathcal{U}(\sigma) \sum_{t=1}^N \log p(x_{\sigma(t)}|\boldsymbol{x}_{\sigma(<t)};\sigma) = \mathbb{E}_{\sigma \sim \mathcal{U}(\sigma)} \sum_{t=1}^N \log p(x_{\sigma(t)}|\boldsymbol{x}_{\sigma(<t)};\sigma)$$

Instead our proposed method uses a delta prior $w(e)$ over orderings (different for each mask), as induced by the protocol $w$-MAC. Due to the delta prior, there is no ELBO gap when using deterministic ordering functions.

$$\log p(\boldsymbol{x}_e) = \log \sum_\sigma p(\sigma)p(\boldsymbol{x}_e|\sigma) = \log \sum_\sigma \mathbb{1}_{\sigma=w(e)} \prod_{t=1}^{|e|} p(x_{\sigma(t)}|\boldsymbol{x}_{\sigma(<t)};\sigma)$$

$$= \log \left[ \prod_{t=1}^{|e|} p(x_{\sigma(t)}|\boldsymbol{x}_{\sigma(<t)};\sigma) \right]_{\sigma=w(e)} = \left[ \sum_{t=1}^{|e|} \log p(x_{\sigma(t)}|\boldsymbol{x}_{\sigma(<t)};\sigma) \right]_{\sigma=w(e)}$$

Although this connection to variable ordering is interesting, we note that deterministic orderings do not fully capture the subtleties of mask distributions. In particular, our decomposition protocol was chosen to increase concentration of intermediate masks (lower entropy), which corresponds to picking variable orderings that are not just deterministic, but that overlap in paths on the lattice.

# B Batch Sampling from MAC's Mask Distribution

Sampling masks from the training distribution *in batch* is important for efficient training. Since MAC uses a simple protocol (always decompose by removing the greatest element in the mask), we can sample masks in batch without simulating the decomposition through the lattice. We provide code snippets for this batch sampling procedure in PyTorch.

```python
from torch import arange, multinomial, rand, randint

def sample_test_masks(batch: int, xdim: int):
    sigma = rand(size=(batch, xdim)).argsort(dim=-1)
    t = randint(low=1, high=xdim+1, size=(batch, 1))
    masks = sigma < t
    return masks, t

def sample_train_masks(batch: int, xdim: int):
    test_masks, test_t = sample_test_masks(batch, xdim)

    # sample intermediate prefix by taking random int in [0, test_t)
    batch_arange = arange(xdim).reshape(1, xdim).repeat(batch, 1)
    nonzero_weights = (batch_arange < test_t).float()
    t = multinomial(nonzero_weights, num_samples=1)
```

```
    # double argsort trick to get ranks, but we need:
    # 1. descending=True to order 1s before 0s of the bitmask
    # 2. stable=True to keep the relative ordering between the 1s
    sigma = test_masks.long().sort(descending=True,
                                    stable=True).indices.argsort()
    masks = sigma < t
    return masks, t

def cardinality_reweighting(batch: int, xdim: int, outer_batch=100):
    train_masks, train_t = sample_train_masks(outer_batch*batch, xdim)
    card_weight = (train_t + 1).float()[:,0]
    idx_select = multinomial(card_weight,
                             num_samples=batch, replacement=False)
    return train_masks[idx_select]
```

The function `sample_test_masks` samples in batch from the test mask distribution $M = M_{\text{card-mask}}$, where we first sample a cardinality uniformly random, and then sample a mask with the chosen cardinality uniformly at random. This function is modular – any other test mask distribution $M$ can be dropped in without modifying rest of the code snippet.

The next function `sample_train_masks` samples masks from the training distribution $D_{M,w\text{-MAC}}$ in batch. It takes each test mask $e$ and samples an intermediate mask on the path between $e$ and $\emptyset$ as dictated by the decomposition protocol $w$-MAC. Since our protocol always removes the greatest element in the set, we can actually sample this intermediate mask without simulating the protocol: sort the elements and choose a prefix with length picked uniformly at random.

Finally, the function `cardinality_reweighting` changes the training distribution from $D_{M,w\text{-MAC}}$ to $D_{M,w\text{-MAC}}^{\text{CR}}$ using the cardinality reweighting heuristic. This is done by Sampling-Importance-Resampling, where we sample an excess (e.g., 100x) number of masks, reweight by their cardinality, and resample using these weights.

To train MAC, we sample masks in batch from $D_{M,w\text{-MAC}}^{\text{CR}}$ by calling `cardinality_reweighting`.

## C  Samples

We show (uncurated) masked-conditional samples from MAC, by masking test data and using MAC to fill in the missing dimensions. We repeat this for Text8, CIFAR10, and ImageNet32.

### C.1  Text8

Each snippet displays 3 chunks of 250 characters. The first chunk is an unmasked example from the test set. The second chunk is its masked out version, with underscores denoting missing dimensions. Finally, the last chunk is the mask-conditional sample, using MAC to fill in the missing dimensions.

```
be ejected and hold it there examine the chamber to ensure it is clear
 allow the action to go forward under control push the forward assist
fire the action and close the ejection port cover safety precaution ma
gazine fitted perform an unload if the a

be _j_c__d a__ h__d _t t__re exa_i_e_the __a__er to __sure_it __ c_ea_
 allo__the a__i__ __ g_ _o_war_ u_der __ntrol p__h _h_ f_rw_rd_ass_st
f_re ___ action _nd_clo_e the _jection p__t _o_er _a_ety _r_ca_ti_n m_
g__i__ _i__ed_p____rm_a_ unlo___if_th_ a

be ejected and hold it to re examine the bladder to ensure it is clear
 allow the action to go forward under control push the forward assist
fire the action and close the ejection part cover safety precaution ma
gazine firied perform an unload if the a
```

```
 prophylactic drugs several drugs most of which are also used for trea
tment of malaria can be taken preventatively generally these drugs are
```

```
 taken daily or weekly at a lower dose than would be used for treatmen
t of a person who had actually contracte
```

```
 pr_p_y__cti__dr__s _eve__l drug_ _o_t of__h__h are __so u_ed __r _re_
tment__f mal__i__c__ b_ _a_en _re_enta__vely ge__ra_l_ t_es_ d_ugs_are
 _ake___aily or _ee_ly _t a lo_er dose t__n _o_ld _e_used_f_r _re_tme_
t__f__ _e__on_w____ad_a_tuall___on_ra_te
```

```
 prophylactic drugs several drugs most of which are also used for trea
tment of malaria can be taken preventatively generally these drugs are
 taken daily or weekly at a lower dose than would be used for treatmen
t of a lesion who had actually contracte
```

---

```
 themselves as a versified journal secondly there are cycles of poems
which fall into a regular chronological sequence among the single poem
s evidence that certain themes demanded further expression and develop
ment one cycle announces the theme of mi
```

```
____mse__e_______v_r__f__d__o_____ _____d_y______ ___ ____e_ o__p____
_h_c_ f__l_____ _ _______ ch___o___i_a__s_______ ___ng_th__si__l_ po_m
________c_ _h____er_ai_ ________e_____d __r____ _x__e______a_d_d_v____
_____o_e____l_ a__o_______he t_em_______
```

```
 themselves they vary foad loafing it a day three the number of poems
which follow at a at b is chronological stead up owing the single poem
 syllagics they certail a unprecedented form of expression and develop
ment one could also feel the theme can f
```

---

## C.2 CIFAR10

Each figure displays 3 rows of 18 images. The first row is an unmasked example from the test set. The second row is its masked out version, with grey pixels denoting missing dimensions. Finally, the last chunk is the mask-conditional sample, using MAC to fill in the missing dimensions. For easier visualization, in these examples we align the masked dimensions for the three image channels (RGB). We do not align them during training and evaluation of test log-likelihoods, i.e., the masks could be different for each channel.

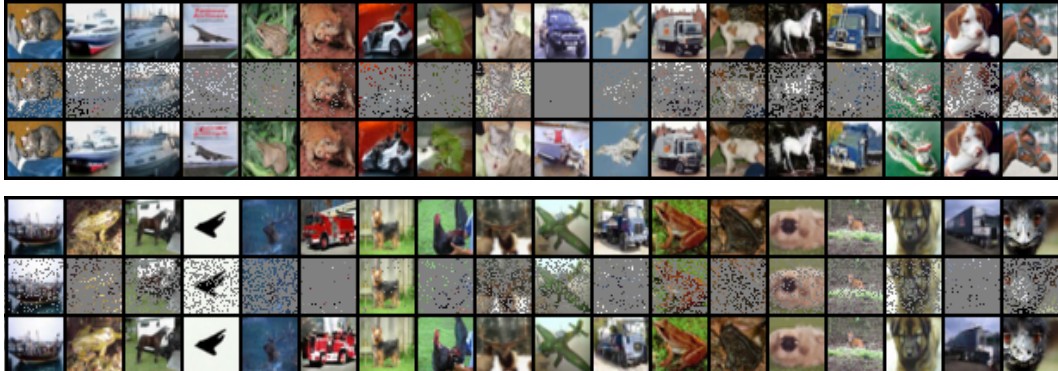

## C.3 ImageNet32

We show samples from ImageNet32, using the same setup as described for CIFAR10 above.

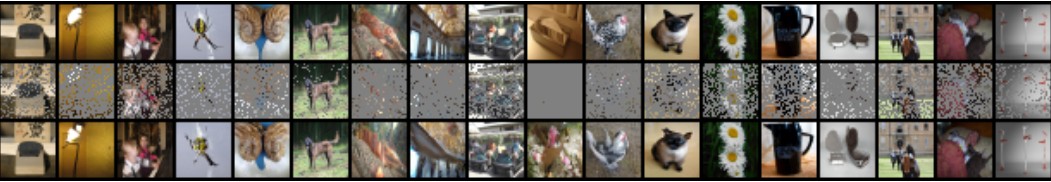

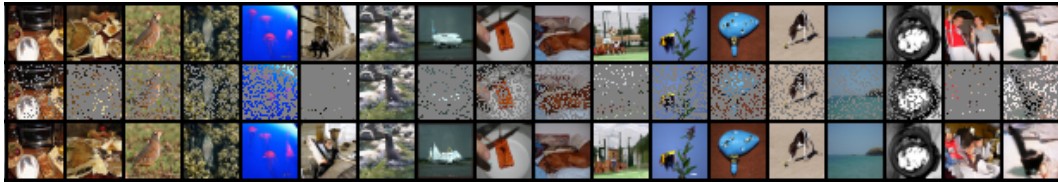

# D   Experimental Details

For the language and image experiments, all of our architectural and hyperparameter choices are kept the same as ARDM [14], with the exception of the Cross-Entropy objective from [1]. We omit this Cross-Entropy objective, as the authors from [14] found "no substantial differences in performance" from including it. Compared to the ARDM baseline, we only modify the training mask distribution and inference protocol.

**Language**

- 12 layer Transformer with 768 dimensions, 12 heads, 3072 MLP dimensions, no dropout
- Batch size 512, chunk size 250 with *no additional context*
- Learning rate 5e-4, with linear warmup for the first 5k steps, using AdamW
- Gradient clipped at 0.25

**Image**

- U-Net with 32 ResBlocks at resolution $32 \times 32$ with 256 channels, interleaved with attention blocks, no dropout
- The mask is concatenated to the input, which is encoded using $3/4$ of the channels. The remaining $1/4$ of the channels encode the mask cardinality (see [14] for details).
- Batch size 128
- Learning rate 1e-4, with beta parameters (0.9 / 0.999), using Adam
- Gradient clipped at 100

**Continuous Tabular Data**   For the continuous tabular data, all of our architectural and hyperparameter choices are kept the same as ACE [35]. Compared to the ACE baseline, we only modify the training mask distribution and inference protocol.

We use fully-connected residual architectures (with 4 residual blocks) for both proposal and energy networks. The proposal network uses a mixture of 10 Gaussian components. The exact configuration for each of the dataset is kept identical to ACE, and can be found in more detail in [35].

**FrankaKitchen**   We train on the `Kitchen-mixed0` task, using a fully connected MLP with width 2048 and depth 8. We optimize using Adam with learning rate 5e-4, for 300 epochs with a batch size of 4k. BC learns an explicit model that directly maps a state to an action. MAC learns an implicit model that models a distribution over the action space given the state, as in [10]. The action distribution is modeled by univariate conditionals parameterized by a mixture of 20 Gaussians.

**Code**   Code for this paper can be found at `https://github.com/AndyShih12/mac`.