# OpenReview forum: "Training and Inference on Any-Order Autoregressive Models the Right Way"
_NeurIPS.cc/2022/Conference — NeurIPS 2022 Accept_

### Official Review · Reviewer_HNCy · 2022-07-07

**Rating:** 7
**Confidence:** 4
**Soundness:** 3 good
**Presentation:** 3 good
**Contribution:** 3 good

**Summary:**

The authors present the Mask-tuned Arbitrary Conditional Models - a method for training Any-order Autoregressive Models.
They argue that the paper presents two insights hurting conventional AO-ARM training which they address with their method.

The authors investigate their method with an ablation study and present it on a variety of experiments.



**Questions:**

I'm not sure what your "sharpen mask distribution" section is concluding. Is it just that your selected mask distribution has lower entropy (by design) than uniform? You're not showing that low entropy mask distributions actually help, except, possibly, in your earlier experiments.

I think there is a small typo in the caption of Figure 1a: "An autoregressive (model?) defines ..."



**Limitations:**

A small discussion on the mask distributional mismatch is included. Generally, limitations, or the lack thereof, are not discussed.

**Strengths And Weaknesses:**

Strengths:

The explanations in the paper are well done and overall I found the paper clear and well-written.

Ablation study well isolates the components of your method and the contribution of each of them.

Experiments are clear and well presented. A good variety of experiments were presented.

The authors state that it is not arbitrary joint distribution decomposition ordering that we really care about but, instead, arbitrary conditioning. They show that an ordering over every marginal can be specified, or a distribution over orderings, to address redundancy in training.

The authors show that tuning the mask distribution for training can lead to better results - an important idea that can be investigated further. Although the idea is straight-forward, I view this as a strength of the method as it is a simple, effective modification to the standard training techniques that yields better results with, as the authors put it, no compromises.

Weaknesses:
A lack of error measurements in most of the experiments makes it hard to determine if your results are significant.

The idea that we want to weight masks more often used (which are the lower carnality ones) and the cardinality reweighting seem contradictory so I'm not sure how much I believe your intuitive explanations of each idea. The ablation study shows the effect of using the insights in the order of B->A but I wonder how well A only would do. Did you try to use a uniform distribution over the masks remaining after insight A (pictured in figure 2c)?

If I understand correctly, your algorithm box doesn't actually reflect how you train the model (in parallel) using the objective in eq 7. There is a lot of machinery presented (although interesting) to arrive at the conclusion that the MAC objective (eq 7) is the standard AO-ARM objective (eq 4) with a different mask sampling distribution.

A subjective point: I found the notation confusing in some places. You use several different types of notation to refer to the same objects - you refer to the conditional distribution of interest as p(x_sigma(t)| x_sigma(<t)), p(x_i_t|x_e_t), p(x_i|x_e), p(x_j| x_e/j). I was particularly confused by referring to edges as (i, e) or (i_t, e_t). What is i in this case? I assume from context it's a variable not contained in the mask e due to the summation on line 207 but I don't think it was defined.

---

> ### Author Response · Authors · 2022-08-02
> **Response**
>
> > **A lack of error measurements in most of the experiments makes it hard to determine if your results are significant.**
>
> Due to computational limitations we were only able to run multiple seeds for the shared autonomy task. However, we believe that the ablation study and the range of different domains is good evidence that our results are significant and reproducible.
>
> &nbsp;
>
> > **The ablation study shows the effect of using the insights in the order of B->A but I wonder how well A only would do. Did you try to use a uniform distribution over the masks remaining after insight A (pictured in figure 2c)?**
>
> Thanks for the suggestion. We had not tried this, but have now added the ablation. Please see the updated Figure 3 with the purple line, which uses only insight A. It gives slight improvements over the baseline, but slightly less than using both insights A and B.
>
> &nbsp;
>
> > **The idea that we want to weight masks more often used (which are the lower cardinality ones) and the cardinality reweighting seem contradictory so I'm not sure how much I believe your intuitive explanations of each idea.**
>
> This is an interesting point of discussion, and one that we also do not fully understand. Here is our current hypothesis of why CR gives consistent improvements.
>
> The ultimate goal is to perform well on masks in the test mask distribution. The machinery of MAC allows us to eliminate many edges, so that the test mask distribution has a smaller support, has lower entropy, and in general makes the learning problem easier. MAC then aligns the training mask distribution to the test mask distribution.
>
> However, the issue is that there are only $n$ masks of cardinality 1, and there are exponentially many masks of ‘medium’ cardinality. Even though masks of cardinality 1 should be sampled very often, there are so few of them that the model can learn them very quickly. Hence, more effort should be devoted to ‘medium’ cardinality masks, of which there are many and are harder for the model to learn.
>
> In summary, it’s clear that we want to eliminate unused masks from our training distribution, and that we should roughly try to match the training and the test mask distribution. It’s less clear that we want to match the mask distribution perfectly, since repetitively sampling the super common masks can lead to ‘diminishing returns’. Again, we don’t claim to fully understand this, but this is our current hypothesis. Empirically at least, CR seems to help consistently, and it may be interesting to explore other reweighting strategies that might help even more.
>
> &nbsp;
>
> > **If I understand correctly, your algorithm box doesn't actually reflect how you train the model (in parallel) using the objective in eq 7**
>
> Yes, the algorithm box reflects the training procedure without the parallelization, which we omitted for clarity. We can include it in the updated version if you wish.
>
> &nbsp;
>
> > **I found the notation confusing in some places. You use several different types of notation to refer to the same objects - you refer to the conditional distribution of interest as p(x_sigma(t)| x_sigma(<t)), p(x_i_t|x_e_t), p(x_i|x_e), p(x_j| x_e/j).**
>
> Sorry for the confusion. We used different notations based on what was most convenient in the context. For example, we use $p(x_\sigma(t)| x_\sigma(<t))$ to describe prior work, since those operated on orderings $\sigma$. For our work, we mostly reason over masks, so we use $p(x_i|x_e)$ or $p(x_j|x_{e \setminus j})$, depending on whether we are going rightward or leftward on the binary lattice.
>
> &nbsp;
>
> > **I was particularly confused by referring to edges as (i, e) or (i_t, e_t). What is i in this case? I assume from context it's a variable not contained in the mask e due to the summation on line 207 but I don't think it was defined.**
>
> Yes, in general we used $i \in X \setminus e$ to denote a variable not contained in the mask $e$, and we used $j \in e$ to denote a variable contained in the mask $e$. We have clarified this in the updated version of the paper.
>
> &nbsp;
>
> > **I'm not sure what your "sharpen mask distribution" section is concluding. Is it just that your selected mask distribution has lower entropy (by design) than uniform? You're not showing that low entropy mask distributions actually help, except, possibly, in your earlier experiments.**
>
> We wanted to use Table 7 to give intuition, in a concrete small-scale setting, on the difference between our distribution and the baseline distribution. In general we think that lower entropy mask distributions should help, but this is not the full story since CR increases entropy but also improves performance (as we discussed earlier).
>
> &nbsp;
>
> > **I think there is a small typo in the caption of Figure 1a: "An autoregressive (model?) defines ..."**
>
> Thanks, we have fixed this.

---

> > ### Comment · Reviewer_HNCy · 2022-08-03
> > **Re: Response**
> >
> > Thanks for the thorough response, clarifications and answers to my questions. I'm glad that you were able to include the new ablation study in the revision. It's too bad about the computational constraints restricting error measurements but I agree with your comment that your paper has provided good evidence that your results are significant.

---

### Official Review · Reviewer_fwZi · 2022-07-11

**Rating:** 8
**Confidence:** 3
**Soundness:** 4 excellent
**Presentation:** 4 excellent
**Contribution:** 4 excellent

**Summary:**

The paper improves Any-order ARO models by  making them more efficient reducing redundancy in computations, while maintaining tractability and by better aligning the training and inference objectives.

The key idea is to reformulate the objectives in AO-ARO such that redundancies in computing the univariate conditionals can be fully exploited. Further, by tuning the training based on the distribution that represents the importance of univariate conditionals, the training is better suited to compute marginals during inference.

Experiments are performed on several tasks and comparison studies show the improvements the proposed approach offers compared to state of the art

**Questions:**

None

**Limitations:**

They are not explicitly addressed. It would be nice to have a discussion on these in the paper.

**Strengths And Weaknesses:**

1. The paper is written very well clearly defining the problem, the background work and its own novel contribution.
2. The work is meaningful and significant considering the generality of computing marginal probabilities (by integrating from high-dimensional data)
3. The experiments seem to cover many areas, standard benchmarks from computer vision, language models, robotics, etc. The variety of experiments clearly shows the strengths of this work.

Weakness

While this may not be a weakness, maybe limitations of the proposed work can be mentioned or discussed.

Overall, I thought the paper proposed a significant problem, clearly outlined a good solution and conducted an excellent empirical study.

---

> ### Author Response · Authors · 2022-08-02
> **Response**
>
> > **maybe limitations of the proposed work can be mentioned or discussed.**
>
> Thank you for your review. We have added some discussion of limitations in the new Related Work and Limitations section. We discuss possibilities of learning the canonical ordering, and the limitations of the approximate nature of AO-ARM models in general.

---

### Official Review · Reviewer_9KgX · 2022-07-11

**Rating:** 6
**Confidence:** 4
**Soundness:** 4 excellent
**Presentation:** 2 fair
**Contribution:** 3 good

**Summary:**

A method for training nonautoregressive generative models of high-dimensional (e.g., sequence) data is proposed. The main algorithm requires the choice of a decomposition protocol, which is a choice of generation order defined by a deterministic Markovian reverse generation process. The training objective is equivalent to a weighted log-likelihood of each generation step in the chosen order leading to a randomly masked training sample. Strong results are shown on a variety of domains.

**Questions:**

- Would the results change if "largest" is replaced by "smallest" (or largest w.r.t. some random but fixed permutation of the dimensions) in the definition of the decomposition protocol (L179)? The definition relies upon a choice of natural ordering in the data space, so it may be important.
   - In particular, how could the algorithm be combined with the learning of a (possibly probabilistic) decomposition protocol?
- Relation to past work.
   - A few papers that have addressed the question of non-autoregressive modeling have not been mentioned. For instance, [A1] is a relevant citation, while [A3] suggests an alternative approach that uses a continuous relaxation to learn generation order.
   - (Also related to the first question:) There are several other algorithms that feature a learned generation order, albeit in slightly different settings. In [A2], non-uniform generation order emerges implicitly from a greedy optimization procedure. In [A4], which uses a similar framing of chains on a lattice, the learned "backward policy" is a stochastic, data-dependent decomposition protocol. What is the relationship of MAC with such approaches?

[A1] B.Uria et al., Neural autoregressive distribution estimation (JMLR, 2016). arXiv:1605.02226

[A2] D.Emelianenko et al., Sequence modeling with unconstrained generation order (NeurIPS 2019). arXiv:1911.00176

[A3] X.Li et al., Discovering non-monotonic autoregressive orderings with variational inference (ICLR 2021). arXiv:2110.15797

[A4] D.Zhang et al., Generative flow networks for discrete probabilistic modeling (ICML 2022). arXiv:2202.01361

**Limitations:**

Please see the second point of Weaknesses above.

**Strengths And Weaknesses:**

Strengths:
- The proposed algorithm is simple and intuitive and the derivation is sound.
- Strong results on a diverse set of problems, including language, vision, tabular data, and robot motion planning. Evaluations and comparisons are solid.

Weaknesses: (I am willing to raise the score if the authors discuss the limitations and relationship with past work in their response and commit to adding them to the paper.)
- The definition of the decomposition protocol used is hidden in the text (L179) and not illustrated on real data. This makes it hard for a reader to check their understanding and gain intuition.
- The use of a canonical ordering of data dimensions seems like a limitation of the approach, but it is not discussed. For example, even in the image domain, it is not clear that the standard ordering of pixels in the "reading" order is optimal for modeling. In other domains, such as tabular data and other settings where incremental generation in arbitrary order is possible (e.g., generic fixed- but high-dimensional data or data of variable size, such as graphs, point clouds, etc.), there is no natural ordering. Please see the questions below.
- The discussion of related work could be improved. Please see the second question below.

---

> ### Author Response · Authors · 2022-08-02
> **Response**
>
> > **The definition of the decomposition protocol used is hidden in the text (L179) and not illustrated on real data. This makes it hard for a reader to check their understanding and gain intuition.**
>
> We’ve highlighted this more in the updated version (end of Section 3).
>
> &nbsp;
>
> > **The use of a canonical ordering of data dimensions…. it is not clear that the standard ordering of pixels in the "reading" order is optimal for modeling… there is no natural ordering. "largest" is replaced by "smallest"?**
>
> We did not explore much the choice of canonical ordering in this paper. Indeed, MAC will work on top of any choice of canonical ordering, since this simply amounts to relabeling nodes on the binary lattice.
>
> We agree that some choices of canonical orderings may lead to better empirical performance. For example, we actually noticed that a strided ordering (0, 32, 64…, 1, 33, 65…) often did better than the lexicographical (0,1,2,3…) order for images. We had left this observation out of the paper since we thought it was out of scope of the main message, but we have now included it in the new Related Work section.
>
> &nbsp;
>
> > **In particular, how could the algorithm be combined with the learning of a (possibly probabilistic) decomposition protocol?**
>
> Learning the canonical ordering / decomposition protocol is an interesting avenue to pursue. As mentioned above, we empirically saw that strided ordering does better than the lexicographical ordering, so using a principled framework for learning the ordering / protocol could lead to promising improvements. We also agree that the citations provided (e.g. A2) would be suitable places to start.
>
> &nbsp;
>
> > **The discussion of related work could be improved. A few papers [A1, A2, A3, A4] that have addressed the question of non-autoregressive modeling have not been mentioned. What is the relationship of MAC with such approaches [A2, A4]?**
>
> Thank you for the suggested references. We have cited these works and discussed their relation in the new Related Work section.
>
> [A2] operates on sets, so the variables are not associated with a particular index, whereas AO-ARMs operate on a joint distribution $(X_1, …, X_n)$ with fixed indices. In other words, in [A2] the main operation is insertion “a cat” -> “a black cat”, whereas for MAC the main operation is unmasking “a [blank] cat” -> “a black cat”. As a result, the interpretation of marginal likelihood is also different in these two settings.
>
> [A4] is also related, using GFlowNets and a reinforcement learning treatment with states, actions and rewards. However it appears that our method scales to larger domains (imagenet32, cifar, text8) versus their approach (MNIST). We suspect this may be because MAC has parallel training, can optimize using MLE instead of energy-based methods, or may be because GFlowNets relies on an RL framework.

---

> > ### Comment · Reviewer_9KgX · 2022-08-03
> > **Response to response**
> >
> > Thank you for the detailed response and the updated discussion. After having seen the changes and read all reviews and responses, I am increasing the rating from 5 to 6.
> >
> > A few thoughts, which are not necessarily points to be addressed in the paper:
> >
> > > We agree that some choices of canonical orderings may lead to better empirical performance. For example, we actually noticed that a strided ordering (0, 32, 64…, 1, 33, 65…) often did better than the lexicographical (0,1,2,3…) order for images. We had left this observation out of the paper since we thought it was out of scope of the main message, but we have now included it in the new Related Work section.
> >
> > This is very interesting. I also wonder about **sample-dependent but fixed** decomposition protocols. For instance, on my first skimming of the paper, I had misread L179 to mean that the variable with the largest *value* is removed, e.g., first generating the dark background and then the foreground of an MNIST digit. This is of course not what was actually done, but it is a possible deterministic protocol that yields different orders of variable indices for different samples.
> >
> > > [A2] operates on sets, so the variables are not associated with a particular index, whereas AO-ARMs operate on a joint distribution  with fixed indices. In other words, in [A2] the main operation is insertion “a cat” -> “a black cat”, whereas for MAC the main operation is unmasking “a [blank] cat” -> “a black cat”. As a result, the interpretation of marginal likelihood is also different in these two settings.
> >
> > This is true, but one can imagine how to adapt the training procedure in [A2] to the setting considered in this paper ("a [blank] cat" -> "a black cat").
> > For *stochastic* decomposition protocols, perhaps a connection can also be made with inverting noising processes, where the transition distribution has an absorbing state [blank], which appears in recent literature on diffusion in discrete spaces.
> >
> > > [A4] is also related, using GFlowNets and a reinforcement learning treatment with states, actions and rewards. However it appears that our method scales to larger domains (imagenet32, cifar, text8) versus their approach (MNIST). We suspect this may be because MAC has parallel training, can optimize using MLE instead of energy-based methods, or may be because GFlowNets relies on an RL framework.
> >
> > Indeed, few scalable algorithms explicitly produce a stochastic decomposition rule. MLE-like models, including the one in this work and those based on denoising, are efficient to train. Could the essential obstacle to combining the two be that parallelism across positions -- like in this paper's equation (7) or in the trick that turns inverting a diffusion SDE into a denoising objective -- is impossible without strong assumptions on the decomposition rule?

---

> > > ### Author Response · Authors · 2022-08-04
> > > **Response**
> > >
> > > > **sample-dependent but fixed decomposition protocol**
> > >
> > > Yes this is a good idea, to determine the ordering on the fly as we uncover pixels. We also agree that this should be possible, and perhaps can be learned as well.
> > >
> > >
> > > > **parallelism across positions is impossible without strong assumptions on the decomposition rule?**
> > >
> > > We do think that techniques such as parallelism need to exploit the right task structures and loss functions, and this can be difficult for frameworks (e.g. GFlowNets) that are designed for generic combinatorial tasks that may not have this structure.

---

### Official Review · Reviewer_2pLm · 2022-07-12

**Rating:** 6
**Confidence:** 3
**Soundness:** 3 good
**Presentation:** 3 good
**Contribution:** 3 good

**Summary:**

The paper argued that one current limitation of Any-Order Autoregressive Models (AO-ARMs) is that they suffer from redundancy in their probabilistic model, i.e., they define the same distribution in multiple different ways. Especially, to my understanding, the authors mean that a sequence of length $n$ can be defined by $n!$ orders, which causes the model to under-fitting any-order data distribution as observed in XLNet or ARDMs. A particular order (i.e., $w$-MAC, Mask-tuned Arbitrary Conditional Model) is introduced to remove such redundancy by sampling the order from a non-uniform decomposition protocol $w$. During the inference (which only contains the likelihood estimation task in this paper), the order is also sampled from $w$-MAC, rather than uniform distribution, and thus achieves better likelihood estimation (somewhere between left-to-right autoregressive model and real AO-ARMs).

**Questions:**

1. Clearly, a biased order distribution can produce better perplexity, as it is trained more on a specific order. And therefore, in principle, it should not perform better than real AO-ARMs, but this is not reflected in the experimental results of Table 1, which caused my confusion. My question is on the evaluation of marginal likelihood. Let's again take Figure 2 (b) as an example, I'm wondering how proposed method can compute the marginal likelihood of $p(x_1, x_2, x_3, x_4 | x_2, x_4)$ if there is no edges between (2, 4) and (1, 2, 3, 4)?

**Limitations:**

The work does not have an obvious potential negative societal impact.

**Strengths And Weaknesses:**

Strengths:

1. The main strength of the paper is that it finds a (maybe principled) way to tweak the order distribution in AO-ARMs to get a better likelihood number (which also might not be true, as I argued in the Weaknesses). This could potentially improve the quality of tasks that require sampling from an AO-ARMs, and in the meantime, can produce a valid probability given any observed and unobserved variables.

2. A heuristic cardinality reweighting trick is introduced in Section 3.2 and explained in Section 5. This may inspire future research in this direction.

3. Despite the high complexity, the model can still be trained in parallel.

Weaknesses:

I mainly have two concerns.

1. Using Figure 2 (b) as an example, in the proposed method, the following equation might hold $p(x_2, x_3, x_4) > p(x_2, x_4)$, which means $\mathbb{E}_{\sigma} p(x_3 | x_2, x_4) > 1$. Based on this, I think the mathematical foundation in the proposed model is invalid.

2. In the evaluation, all the previous work samples a random order $\sigma$ from the uniform distribution because of the following assumption:
   1.  "In other words, it does not matter which step $t + k$ the model predicts, in expectation, these all have the same associated likelihood." from [1]
   2.  "We note that a model which perfectly captures the true densities would give consistent likelihoods for all possible orderings (thus evaluating only one ordering would suffice)." AND Algorithm 1 from [2]

However, in this work, $\sigma$ is no longer from a uniform distribution. Therefore, in order to get a precise likelihood estimation, the paper might need to get more order samples to reduce the variance in likelihood estimation. Still, no such information or discussion is provided in the paper.

[1] Autoregressive Diffusion Models
[2] Arbitrary Conditional Distributions with Energy

---

> ### Author Response · Authors · 2022-08-02
> **Response**
>
> To clarify terminology, we refer to marginals as quantities such as $p(x2, x4)$, and conditionals as quantities such as $p(x1, x3 | x2, x4)$.
>
> &nbsp;
>
> > **Using Figure 2 (b) as an example, in the proposed method, the following equation might hold $p(x2,x3,x4)>p(x2,x4)$, which means $\mathbb{E}_{\sigma} p(x3|x2,x4)>1$. Based on this, I think the mathematical foundation in the proposed model is invalid.**
>
> There are two points on this.
>
> - Indeed, evaluating conditionals as two marginals can be problematic, and this is a limitation of both standard AO-ARMs and MAC. Even in standard AO-ARMs, it could be the case that $p(x2,x3,x4)>p(x2,x4)$ (easy to construct an example if you wish). This is because marginal likelihoods are only evaluated using compatible orderings, not using all possible orders.
> \
> \
> In fact, this means marginal and conditional estimates for both AO-ARM / MAC in general can be biased.
>
> - Despite the bias, we can still ensure that the conditional probability estimate is valid by evaluating the conditional directly (and not as two marginals). We can do this for MAC as well. Figure 2 showed a version of MAC that only optimized for marginal likelihoods. In Section 4.3 and Tables 4 & 5, we used a version of MAC that optimized for both marginal and conditional likelihoods by training on the corresponding mask distribution.
> \
> \
> To give some intuition on conditional likelihood queries, these correspond to paths between two nodes on the lattice. For these, MAC also provides an important improvement over baseline AO-ARMs. MAC will focus training on a single path between two nodes, whereas AO-ARMs will uniformly sample paths between two nodes.
>
> We’ve added these discussions to the Limitation section in the updated version.
>
> &nbsp;
>
> > **The paper might need to get more order samples to reduce the variance in likelihood estimation. Still, no such information or discussion is provided in the paper.**
>
> Actually, it is the opposite. Our method essentially has no variance due to the deterministic ordering protocol, whereas standard AO-ARMs require more samples to reduce their variance (in practice, as you said, they just take a single sample). The discussion in Appendix A may also be relevant.
>
> &nbsp;
>
> > **Clearly, a biased order distribution can produce better perplexity, as it is trained more on a specific order. And therefore, in principle, it should not perform better than real AO-ARMs, but this is not reflected in the experimental results of Table 1, which caused my confusion.**
>
> Our method uses a biased order distribution, so it produces better perplexity, as you said. So, our method should perform better than standard AO-ARMs. This is reflected in Table1, where lower is better, and at 3000 epochs our method gives 1.40 and AO-ARM gives 1.48.
>
> &nbsp;
>
> > **My question is on the evaluation of marginal likelihood. Let's again take Figure 2 (b) as an example, I'm wondering how proposed method can compute the marginal likelihood of $p(x1,x2,x3,x4|x2,x4)$ if there is no edges between (2, 4) and (1, 2, 3, 4)?**
>
> We write $p(x1,x2,x3,x4|x2,x4)$ as $p(x1,x3|x2,x4)$, using the notation in the paper. This is a conditional likelihood estimate, whereas the diagram in Figure 2b focused on marginal likelihood estimates. To handle a mix of both marginal and conditional likelihood estimates (e.g. Section 4.3) we use a training distribution that matches the corresponding mixed test mask distribution. Then, we can evaluate $p(x1,x3|x2,x4)=p(x1|x2,x4)p(x3|x1,x2,x4)$, as we did for Table 5. Alternatively, we can also evaluate $p(x1,x2,x3,x4) / p(x2, x4)$ as two marginal likelihood estimates, which comes with pros/cons as discussed above.

---

> > ### Comment · Reviewer_2pLm · 2022-08-04
> > **Thanks for the response**
> >
> > I thank the authors for their response and explanations. However, I still have some confusion on the probability model introduced in the paper.
> >
> > > Alternatively, we can also evaluate $p(x1, x2, x3, x4)/p(x2,x4)$ as two marginal likelihood estimates, which comes with pros/cons as discussed above.
> >
> > As discussed in the updated paper, this can lead to invalid conditional probability estimates where $p(x1, x3)/p(x2,x4) > 1$.
> >
> > > Then, we can evaluate $p(x1, x3|x2, x4) = p(x1|x2, x4)p(x3|x1,x2,x4)$, as we did for Table 5
> >
> > I'm still confused about this equation. In Figure 2 (b) or (c), there is no edge between (x2, x4) and x1, or (x1, x2, x4) and x3. Since both p(x1|x2, x4) and p(x3|x1,x2,x4) are never seen by the model in the training, how does AO-ARM estimate these two conditional probability?

---

> > > ### Author Response · Authors · 2022-08-04
> > > **Response**
> > >
> > > > **In Figure 2 (b) or (c), there is no edge between (x2, x4) and x1, or (x1, x2, x4) and x3. Since both p(x1|x2, x4) and p(x3|x1,x2,x4) are never seen by the model in the training, how does AO-ARM estimate these two conditional probability?**
> > >
> > >
> > > For the experiments in Section 4.3, where we do compute conditional probabilities, we trained on a different mask distribution (not shown in Figure 2). This mask distribution (call it Q) corresponds to both marginal and conditional queries, so *all the edges are seen by the model during training*.
> > > Everything else about the training procedure, architecture, inference, etc... remains the same. In other words, to train on Q, we sample queries (which can now be either marginals or conditionals) and apply the decomposition protocol to get a mask, and train on outgoing edges of that mask in parallel. Also, because Q is induced by the decomposition protocol, Q is different from the standard AO-ARM (uniform) distribution, so we do see improvements over the baselines.

---

> > > > ### Comment · Reviewer_2pLm · 2022-08-05
> > > > **Thank you for the explanation**
> > > >
> > > > I thank the authors for the detailed explanation, which solves my main confusion. I am happy to increase my score to 6 to support the accept of the paper.

---

### Author Response · Authors · 2022-08-02
**Response to All**

Thank you all for the detailed reviews of our paper. We are happy to hear that the reviewers found the paper “clear and well-written”, with a “simple, intuitive” and “sound” approach and “strong results on a diverse set of problems”. We are also grateful for all the constructive feedback for improving the paper, and have updated the pdf with the new edits, highlighted in red.

Here is a summary of the edits to the pdf:

- A Related Work section discussing other non-autoregressive sequence modeling work, and possibilities of learning a decomposition protocol. [For Reviewer 9KgX]
- A Limitation section discussing the limitation of the approximate nature of AO-ARMs and MAC model. In short, marginal/conditional estimates may be biased (w.r.t. joint likelihood) but are still valid. [For Reviewer 2pLm and fwZi]
- Updating the ablation study to include one that just uses insight A. As expected it does slightly better than baseline and slightly worse than using both insights A & B. [For Reviewer HNCy]
- Clarifying the decomposition protocol and some of the notation for indexing. [For Reviewer 9KgX and HNCy]

---

### Meta-Review · Area_Chair_eaqH · 2022-08-22

**Recommendation:** Accept
**Confidence:** Certain

**Metareview:**

This paper introduces an improved training method for auto-regressive generative models. Specifically, the paper identifies a redundancy problem in common auto-regressive models and proposes a way to fix this. The reviewers found the contribution significant and important, and it is likely that the paper will have substantial impact.



**Award:**

No

---

### Decision · Program_Chairs · 2022-09-14

Accept